# High numbers of COVID-19 patients transit through non-COVID wards, and associated healthcare workers have high infection rates: An observational cross-sectional study

**Susanna Nallamilli**[1], **Tejus Patel**[1], **April Buazon**[2], **Jennifer Vidler**[3], **Sam Norton**[4], **Mustafa Atta**[2], **James Galloway**[4], **Stella Bowcock**[3]*

1 Department of Medicine, Princess Royal University Hospital, King's College Hospital NHS Trust, Kent, Orpington, United Kingdom, 2 Department of Microbiology, Princess Royal University Hospital, King's College Hospital NHS Trust, Kent, Orpington, United Kingdom, 3 Department of Haematology, Princess Royal University Hospital, King's College Hospital NHS Trust, Kent, Orpington, United Kingdom, 4 Centre for Rheumatic Disease, King's College London, London, United Kingdom

* stella.bowcock@nhs.net, stellashee@yahoo.co.uk

**Data Availability Statement:** All relevant data are within the paper and its Supporting Information files.

## Abstract

Infection risk is high in healthcare workers working with COVID-19 patients but the risk in non-COVID clinical environments is less clear. We measured infection rates early in the pandemic by SARS-CoV-2 antibody and/or a positive PCR test in 1118 HCWs within various hospital environments with particular focus on non-COVID clinical areas. Infection risk on non-COVID wards was estimated through the surrogate metric of numbers of patients transferred from a non-COVID to a COVID ward. Staff infection rates increased with likelihood of COVID exposure and suggested high risk in non-COVID clinical areas (non patient-facing 23.2% versus patient-facing in either non-COVID environments 31.5% or COVID wards 44%). High numbers of patients admitted to COVID wards had initially been admitted to designated non-COVID wards (22–48% at peak). Infection risk was high during a pandemic in all clinical environments and non-COVID designation may provide false reassurance. Our findings support the need for common personal protective equipment standards in all clinical areas, irrespective of COVID/non-COVID designation.

## Introduction

The severe acute respiratory syndrome coronavirus 2 (SARS-CoV-2) swept the world rapidly in early 2020, before we had fully understood the transmission characteristics and illness spectrum caused by the virus [1]. Early in the pandemic, testing capacity for the virus was limited, turnaround slow and the test was insufficiently sensitive to enable effective separation of patients at the point of admission [2]. The separation of patients into non-COVID and COVID areas was, therefore, inevitably imperfect. London experienced an early wave of infections, with cases appearing in late February 2020 and rising steeply over the following 3 months [3]. Healthcare workers (HCWs) working with COVID-19 patients have a higher

**Funding:** The author(s) received no specific funding for this work.

**Competing interests:** The authors have declared that no competing interests exist.

infection risk than those with no patient contact [4, 5]. It is less clear whether HCWs working in non-COVID clinical environments were also at increased risk of infection. Recommendations from Public Health England (PHE) specified that HCWs working on COVID wards should wear PPE. There was no requirement to wear personal protective equipment (PPE) on non-COVID wards. We aimed to assess HCW SARS-CoV-2 infection during the initial pandemic in an acute care facility in London, with particular reference to non-COVID clinical environments. The facility has 454 acute inpatient and 120 elective and rehabilitation beds with approximately 1:4 distribution of single rooms to open 4-beded bays.

## Methods

### Study concept

All hospital staff were invited for a voluntary blood test between May 25th-July 10th 2020, if they wished to know their SARS-CoV-2 antibody status. Blood testing was done under the national NHS England & Improvement voluntary healthcare staff testing programme and took place independently of this audit. Our study was conducted as an internal hospital service evaluation during the blood testing.

### Subjects and site of study

On attending the central hospital blood testing facility, staff were given an information sheet on the project, gave signed informed consent and were asked to complete a voluntary questionnaire detailing in which areas of the hospital they had worked between 17th February-25th May 2020, age group, ethnicity, COVID-19 patient exposure, days of sickness, SARS-CoV-2 PCR swab test results, WHO COVID-19 defined symptoms experienced during this period, symptoms in household contacts, modes of transport utilised and suggestions for improvement. The questionnaire can be found in S1 Table. All hospital staff who attended the phlebotomy service and completed a questionnaire, with consent were eligible. Questionnaires were available throughout the testing period. The voluntary nature of the blood testing and completion of the questionnaire meant the study sample may not have been fully representative of all hospital staff. Demographics of the staff recruited are shown in Table 1 and in S2 Table.

### Testing

Blood samples were screened at a UKAS accredited laboratory for antibodies against SARS-CoV-2 spike protein using the Fortress COVID-19 Total Antibody ELISA kit. Nasopharyngeal swabs were tested using RealStar® SARS-CoV-2 RT-PCR2 RU Kit (Altona diagnostics). Evidence of SARS-CoV-2 infection was measured by the presence of positive antibodies and/or a positive PCR test.

### Exposure risk

Staff were allocated a risk exposure according to their detailed working environment and grouped into non-patient facing and patient-facing environments. The latter were further subcategorised into solely non-COVID environments, mixed exposure or COVID wards. SARS-CoV-2 infection risk in non-COVID wards was measured by the numbers of patients transferred from a non-COVID to a COVID ward. The total number of weekly hospital inpatient admissions and numbers of those who were treated as COVID positive on admission were recorded.

**Table 1. Demographics of all 1118 staff by evidence of infection (positive serology or positive PCR versus negative for both tests).** Values are all n (%) with p-values from chi-square tests (see S2 Table).

| Variable | | Total | negative/indeterminate | positive |
|---|---|---|---|---|
| | | N = 1,118 (%) | N = 733 (65.6%) | N = 385 (34.4%) |
| Age (years) | 18–30 | 295 | 191 (64.7) | 104 (35.3) |
| | 31–40 | 256 | 164 (64.1) | 92 (35.9) |
| | 41–50 | 260 | 163 (62.7) | 97 (37.3) |
| | 51–60 | 232 | 154 (66.4) | 78 (33.6) |
| | >60 | 71 | 57 (80.3) | 14 (19.7) |
| | unknown | 4 | 4 (100.0) | 0 (0.0) |
| Gender | male | 233 | 151 (64.8) | 82 (35.2) |
| | female | 878 | 575 (65.5) | 303 (34.5) |
| | unknown | 7 | 7 (100.0) | 0 (0.0) |
| Ethnicity | white | 592 | 440 (74.3) | 152 (25.7) |
| | black | 118 | 64 (54.2) | 54 (45.8) |
| | asian | 315 | 171 (54.3) | 144 (45.7) |
| | mixed | 24 | 17 (70.8) | 7 (29.2) |
| | other | 37 | 19 (51.4) | 18 (48.6) |
| | unknown | 32 | 22 (68.8) | 10 (31.2) |
| BAME | white/unknown | 624 | 462 (74.0) | 162 (26.0) |
| | BAME | 494 | 271 (54.9) | 223 (45.1) |
| Public transport | no | 740 | 508 (68.6) | 232 (31.4) |
| | yes | 378 | 225 (59.5) | 153 (40.5) |
| Risk allocation | Laboratory (NPF) | 74 | 66 (89.2) | 8 (10.8) |
| | Non-clinical hospital staff (NPF) | 220 | 169 (76.8) | 51 (23.2) |
| | Non-COVID wards only (PF) | 286 | 196 (68.5) | 90 (31.5) |
| | Mixed exposure (PF) | 346 | 194 (56.1) | 152 (43.9) |
| | COVID wards throughout (PF) | 168 | 94 (56.0) | 74 (44.0) |
| | Patient facing—Unknown | 24 | 14 (58.3) | 10 (41.7) |
| Symptomatic household contacts | | | | |
| | No | 718 | 517 (72.0) | 201 (28.0) |
| | Yes | 350 | 185 (52.9) | 165 (47.1) |
| | Unknown | 50 | 31 (62.0) | 19 (38.0) |

NPF–non patient-facing, PF–patient-facing, BAME: Black and minority ethnic

## Data analysis

The data were pseudonymised for analysis. Multiple binary logistic regression models were used to estimate odds ratio's for positive serology between exposure risk categories and prevalence rates adjusted for potential confounders. Two models were estimated with exposure risk entered as a predictor with different categorisations: i) broadly categorising staff as patient-facing versus non patient-facing (binary variable); ii) more granular categories to separate patient-facing staff by time spent on COVID wards (ordinal variable with 4-levels). Age, gender, ethnicity, mode of transport, and positive household contacts were included in each model as dummy coded variables. These variables were considered as potential confounders since these factors had been demonstrated in previous studies to be associated with infection risk and were potentially correlated with staff environment. Estimates from all models are included as supplementary material.

## Ethics

The study was submitted to the hospital management board who approved this service evaluation. Approval for blood testing was under the national NHS England & Improvement voluntary healthcare staff testing programme and was independent of this audit. Written informed consent was obtained to access staff SARS-CoV-2 PCR and serology results.

## Results

1168 (33%) questionnaires were returned during 3520 staff blood tests. Fifty participants were excluded (5 incomplete records, 17 no blood sample, 28 prolonged home/community working). The demographic characteristics of the 1118 staff included are shown in Table 1 divided by the main outcome (evidence of infection or not). Further demographic details are in the S2 Table. Overall, 385 individuals (34.4%; 95%CI 31.7 to 37.3) had evidence of SARS-CoV-2 infection (383 antibody positive and 2 swab positive but antibody negative). Of those with positive antibodies against SARS-CoV-2, 203 (52.7%; 95% CI 47.6 to 57.8%) reported never having had a swab, despite 125 (61.6%; 95%CI 54.5 to 68.3) having experienced symptoms.

The regression analyses showed the following factors associated with an increased risk of infection: Black and minority ethnic (BAME) versus white, household contact with symptoms, and patient exposure risk (Table 2). Staff who were patient-facing versus non patient-facing were more than twice as likely to have evidence of infection (adjusted prevalence 38.6% vs 23.0%; OR 2.19; 95%CI 1.57 to 3.07). Using more granular categorisation demonstrated a

**Table 2. Multiple logistic regression of the risk of infection on selected predictors.**

|  |  | Adjusted odds ratio | 95% confidence interval | p value |
|---|---|---|---|---|
| **Age (years)** |  |  |  |  |
|  | 18–30 | 1.0 |  |  |
|  | 31–40 | 1.10 | 0.763–1.600 | 0.599 |
|  | 41–50 | 1.27 | 0.873–1.841 | 0.213 |
|  | 51–60 | 1.64 | 1.094–2.457 | 0.017 |
|  | > 60 | 0.86 | 0.442–1.672 | 0.657 |
| **Gender** |  |  |  |  |
|  | Male | 1.0 |  |  |
|  | Female | 1.01 | 0.729–1.391 | 0.966 |
| **Ethnicity** |  |  |  |  |
|  | White | 1.0 |  |  |
|  | BAME | 1.89 | 1.422–2.510 | <0.001 |
| **Public transport used** |  |  |  |  |
|  | No | 1.0 |  |  |
|  | Yes | 1.13 | 0.847–1.506 | 0.408 |
| **Household symptoms?** |  |  |  |  |
|  | No | 1.0 |  |  |
|  | Yes | 2.16 | 1.631–2.854 | <0.001 |
|  | Unknown | 1.81 | 0.964–3.407 | 0.065 |
| Risk exposure using non PF as reference | Non patient-facing | 1.0 |  |  |
|  | PF non-COVID | 1.60 | 1.074–2.371 | 0.021 |
|  | PF mixed | 2.63 | 1.812–3.836 | <0.001 |
|  | PF COVID | 2.61 | 1.682–4.063 | <0.001 |

NPF–non patient-facing, PF–patient-facing, BAME: Black and minority ethnic

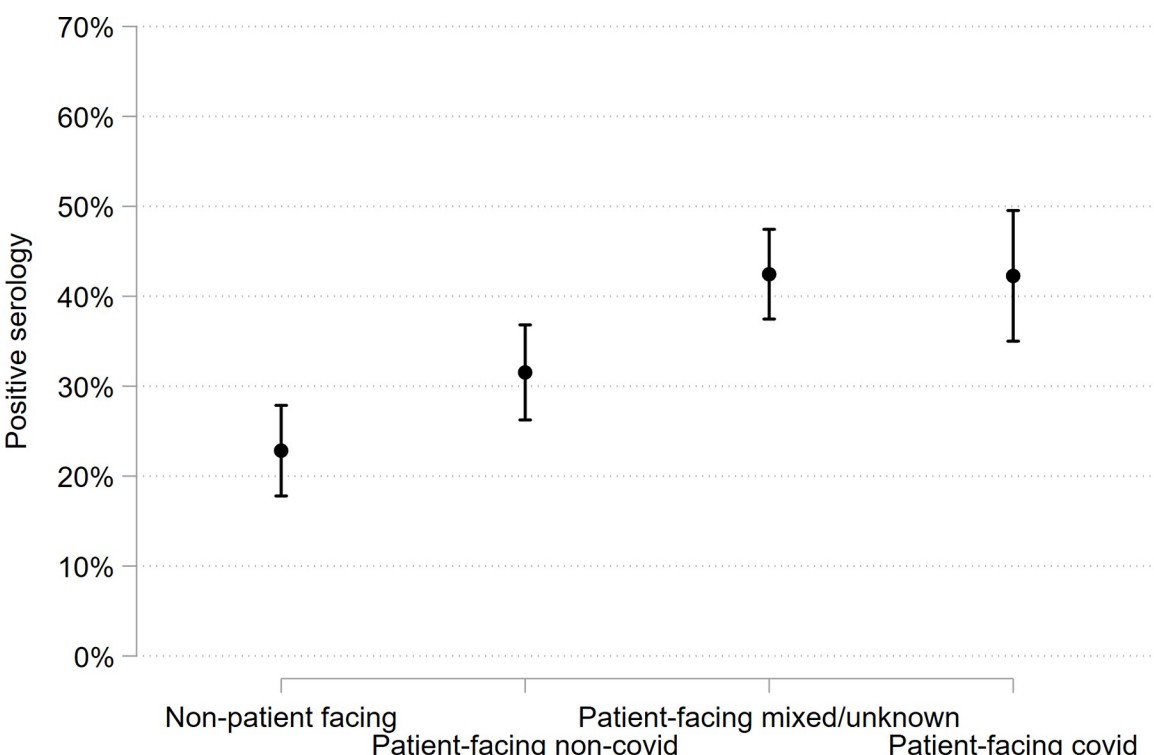

**Fig 1. Adjusted predicted probabilities of infection for the patient facing variable using different patient-facing groupings where these are adjusted for age, gender, BAME, public transport and household contacts.** Non patient-facing cohort compared to all patient-facing subcategories.

stepwise increase by risk of COVID-19 patient exposure (Table 2 & Fig 1). Even those on non-COVID wards throughout had around a one-in-three chance of infection (prevalence 31.5%).

Fig 2 shows the numbers of patients admitted to a COVID ward each week during the peak pandemic period, broken into those by direct admission or those transferred from a

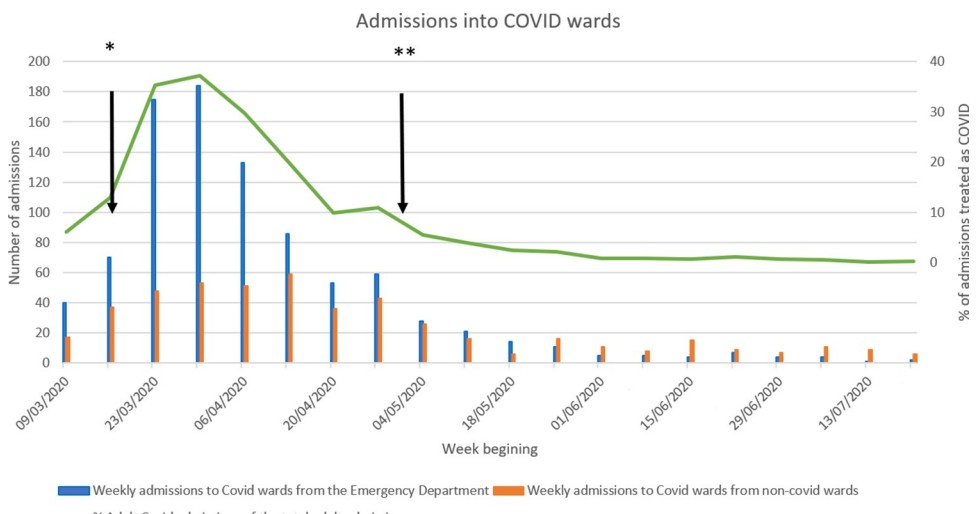

**Fig 2. Breakdown of the routes of admission of adult COVID patients into a COVID ward during the initial pandemic.** * Start of national UK lockdown. ** Mandatory admissions testing commenced.

non-COVID ward, with a reference curve demonstrating the percentage of adult admissions that were initially coded as COVID on admission. A high number of patients on COVID wards during the first peak pandemic had initially been admitted to a non-COVID ward (22–48%). As the wave subsided, non-COVID ward transfers to COVID wards exceeded direct admissions with COVID, though numbers were small. S4 Table summarises staff comments for improvement. There were 234 replies with 261 comments. Main staff concerns were lack of reliable rapid testing, limited isolation facilities, insufficient patient separation and concern about appropriate ward allocation. Suggestions included separate hospital COVID floors and lifts, and less reliance on test results alone for ward allocation. Although staff were not asked about PPE, 9% of replies mentioned inadequacy as a concern.

## Discussion

### Evidence of increased COVID-19 in HCWs

Our data confirm previous observations that HCWs working in patient-facing environments are at increased risk of infection compared to those in non patient-facing environments, [5, 6] and those of BAME background and with symptomatic household contacts are at higher risk (Table 2). Infection risk increased with risk of COVID-19 patient exposure. Our finding of 31.5% infection rate in staff from patient-facing non-COVID environments is similar to that found in another study [7] at 28.9% and contrasts with 23.5% in non patient-facing staff.

### COVID-19 patients on non-COVID wards

There were large numbers of patients who were initially admitted to a non-COVID ward and later transferred to a COVID ward (Fig 2). It is, therefore, not surprising that HCWs in patient-facing non-COVID environments were at increased risk of infection. This observation is supported by other studies that show clusters of infection outbreaks in non-COVID wards [6, 8]. Possibly, the unexpectedness of having unknown COVID status patients in non-COVID areas and reduced use of PPE may have enhanced HCW infection risk. The percentage of patients admitted to COVID wards who originated in non-COVID wards increased with time suggesting there may have been some nosocomial transmission. Nosocomial transmission has been reported to occur [9] but precise measurement is difficult because of overlap between some patients admitted with COVID displaying atypical symptoms, and those who acquired COVID after admission. The staff suggestions for improvement showed considerable concern about the difficulty in discriminating COVID from non-COVID patients and misclassification at the time. A possible way to mitigate this is to improve decisions about patient ward moves, including adding clinical input into decisions as well as requiring negative test results. Initially swab testing capacity for patients and staff was limited as evidenced by how many symptomatic staff had not had a swab test done. Whilst testing is now more widely available and timely than in March-May 2020, tests have a significant false negative rate [2]. Thus, the problem may have diminished but still continues. PPE availability according to PHE guidance was not a significant issue in the trust but some staff were concerned that the recommended level of protection may have been inadequate. Deficiencies detected retrospectively in this study were likely due to lack of knowledge about the virus at the time and testing and environmental constraints rather than poor professionalism. Epidemiological studies show both how infectious and how difficult it is to prevent spread of the virus within closed communities [10, 11]. Similar challenges are likely to have been replicated across the world and this study offers an opportunity to understand the scale of the challenge.

### Strengths and limitations

Strengths of this real world experience study were the large numbers of staff recruited soon after the initial wave of infection, carefully detailed staff categorisation about COVID-19 patient exposure and the high numbers of COVID-19 patients hospitalised. The combination of staff infection rates and detailed ward exposure have rarely been captured.

Limitations include possible sampling bias due to incomplete staff capture: Possibly there was a bias towards staff who regarded themselves at high risk for SARS-CoV-2 exposure. However, the seroprevalence of London blood donors in May 2020 was high at 17.5% [3] and does not contrast too much from the seroprevalence of those in non patient-facing roles at 23.2%. Also, inability to trace source of exposure (no viral fingerprinting) and, therefore, it was impossible to infer causality with certainty. External validity is limited by the fact that the pandemic is evolving, and these findings relate to experiences with a very high community case prevalence. In times of much lower case prevalence, findings would likely be very different.

## Conclusion

Labelling hospital wards as COVID/non-COVID can provide a false sense of security for staff. Our data clearly show that transmission rates amongst staff are substantial even on the non-COVID wards. During periods of high community COVID prevalence and with the emergence of new more infectious SARS-CoV-2 variants, all wards should adhere to common PPE standards if we want to minimise viral transmission in health care facilities.

## Supporting information

**S1 Table. Questionnaire completed by participants.**
(DOCX)

**S2 Table. Demographics of all 1118 staff by evidence of infection (positive serology or positive PCR versus negative for both tests).** Values are all n (%) with p-values from chi-square tests. NPF–non patient-facing, PF–patient-facing.
(DOCX)

**S3 Table. Demographics of all 1118 staff by role (patient-facing or non patient-facing).** Values are all n (%) with p-values from chi-square tests.
(DOCX)

**S4 Table. Themes in staff responses to the question 'do you have any suggestions for ways in which we can improve the processes for separating COVID and non-COVID patients for the future?'.** *Staff were not specifically asked about PPE but these were free-text responses to the above question.
(DOCX)

**S1 Fig. Adjusted predicted probabilities of infection for all risk exposure categories where these are adjusted for age, gender, BAME, public transport and household contacts.**
(DOCX)

**S1 Dataset.**
(PDF)

## Acknowledgments

We thank Nergish Desai, Head of Infection Surveillance and Dominic Thurgood, Business Intelligence Unit, King's College Hospital NHS Trust for their invaluable help with the data.

## Author Contributions

**Conceptualization:** Susanna Nallamilli, April Buazon, Jennifer Vidler, Mustafa Atta, Stella Bowcock.

**Data curation:** Susanna Nallamilli, Tejus Patel, April Buazon, Jennifer Vidler.

**Formal analysis:** Sam Norton.

**Writing – original draft:** Susanna Nallamilli, Jennifer Vidler, James Galloway, Stella Bowcock.

**Writing – review & editing:** Susanna Nallamilli, Tejus Patel, April Buazon, Jennifer Vidler, Sam Norton, Mustafa Atta, James Galloway, Stella Bowcock.

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
