## [Decision Letter · Decision Letter 0]

28 Apr 2021

PONE-D-21-07362

High numbers of COVID-19 patients transit through non-COVID wards, and associated healthcare workers have high infection rates: An observational cross-sectional study

PLOS ONE

Dear Dr. Bowcock,

Thank you for submitting your manuscript to PLOS ONE. After careful consideration, we feel that it has merit but does not fully meet PLOS ONE’s publication criteria as it currently stands. Therefore, we invite you to submit a revised version of the manuscript that addresses the points raised during the review process.

We look forward to receiving your revised manuscript.

Kind regards,

Prasenjit Mitra, MD, MRSB, MIScT, FACSc, FAACC

Academic Editor

PLOS ONE

Journal Requirements:

3. Please provide additional details regarding participant consent. In the ethics statement in the Methods and online submission information, please ensure that you have specified whether consent was informed.

4. Please include additional information regarding the survey or questionnaire used in the study and ensure that you have provided sufficient details that others could replicate the analyses. For instance, if you developed the survey or questionnaire as part of this study and it is not under a copyright more restrictive than CC-BY, please include a copy, in both the original language and English, as Supporting Information. If the questionnaire is published, please provide a citation to the (1) questionnaire and/or (2) original publication associated with the questionnaire.

5. In your Methods section, please provide additional information about the participant recruitment method and the demographic details of your participants. Please ensure you have provided sufficient details to replicate the analyses such as:

a) the place where participants were recruited,

b) a description of any inclusion/exclusion criteria that were applied to participant recruitment,

c) a table of relevant demographic details,

d) a statement as to whether your sample can be considered representative of a larger population.

6. We note that you have indicated that data from this study are available upon request. PLOS only allows data to be available upon request if there are legal or ethical restrictions on sharing data publicly. For information on unacceptable data access restrictions, please see http://journals.plos.org/plosone/s/data-availability#loc-unacceptable-data-access-restrictions.

7. Please upload a copy of Figure 2, to which you refer in your text on page 2 and 3. If the figure is no longer to be included as part of the submission please remove all reference to it within the text.

8. Please include a caption for figure 1 and 2.

9. Please include a copy of Table 1 which you refer to in your text on page 2.

10. Please include a copy of Table 2 which you refer to in your text on page 2.

11. Please include a copy of Table 3 which you refer to in your text on page 3.

12. Please include captions for your Supporting Information files at the end of your manuscript, and update any in-text citations to match accordingly. Please see our Supporting Information guidelines for more information: http://journals.plos.org/plosone/s/supporting-information.

Reviewers' comments:

Reviewer's Responses to Questions

**Comments to the Author**

1. Is the manuscript technically sound, and do the data support the conclusions?

Reviewer #1: Yes

2. Has the statistical analysis been performed appropriately and rigorously? 

Reviewer #1: Yes

3. Have the authors made all data underlying the findings in their manuscript fully available?

Reviewer #1: Yes

4. Is the manuscript presented in an intelligible fashion and written in standard English?

Reviewer #1: Yes

5. Review Comments to the Author

Reviewer #1: This is an interesting manuscript showing that the risk of infection in Non-COVID wards is high for both non-patient and patient-facing staff although it is higher in the latter group. Hence, PPE should be warn by all staff in COVID and non-COVID wards.

There are a few punctuation omissions, such as commas before and after 'therefore' in the middle of a sentence. 'Possibly' at the beginning of a sentence should have a comma. Turnaround should be one word.

6. PLOS authors have the option to publish the peer review history of their article (what does this mean?). If published, this will include your full peer review and any attached files.

Reviewer #1: No

---

## [Author Response · Author response to Decision Letter 0]

13 Jun 2021

Thank you to the reviewer for their comments, and for consideration of this article for publication. Please find the response to the comments below.

Response: We have checked and edited the manuscript as per the style requirements

Response: We have updated the reference list to reflect the most recent citation for each reference. 

3. Please provide additional details regarding participant consent. In the ethics statement in the Methods and online submission information, please ensure that you have specified whether consent was informed.

Response: Please find further information now added in lines 50-53

4. Please include additional information regarding the survey or questionnaire used in the study and ensure that you have provided sufficient details that others could replicate the analyses. For instance, if you developed the survey or questionnaire as part of this study and it is not under a copyright more restrictive than CC-BY, please include a copy, in both the original language and English, as Supporting Information. If the questionnaire is published, please provide a citation to the (1) questionnaire and/or (2) original publication associated with the questionnaire.

Response: Please find the questionnaire in the new supplementary table 1 in our ‘supporting information’ section. The questionnaire has not been published. 

5. In your Methods section, please provide additional information about the participant recruitment method and the demographic details of your participants. Please ensure you have provided sufficient details to replicate the analyses such as:

a) the place where participants were recruited,

b) a description of any inclusion/exclusion criteria that were applied to participant recruitment,

c) a table of relevant demographic details,

d) a statement as to whether your sample can be considered representative of a larger population.

Response: Please find added information regarding this in the methods section, lines 50-56. Supplementary table 2 lists the demographic details and can be found in the ‘supporting information’ document uploaded.

6. We note that you have indicated that data from this study are available upon request. PLOS only allows data to be available upon request if there are legal or ethical restrictions on sharing data publicly.

Response: As there are no ethical or legal restrictions on sharing this data set, please find an anonymised data set uploaded onto the submission portal. 

7. Please upload a copy of Figure 2, to which you refer in your text on page 2 and 3. If the figure is no longer to be included as part of the submission please remove all reference to it within the text.

Response: Please find the figure uploaded onto the submission portal

8. Please include a caption for figure 1 and 2.

Response: Please find the captions for both figures within the manuscript, just below the paragraph in which they are initially referenced

9. Please include a copy of Table 1 which you refer to in your text on page 2.

10. Please include a copy of Table 2 which you refer to in your text on page 2.

Response: Please find both tables within the manuscript text

11. Please include a copy of Table 3 which you refer to in your text on page 3.

Response: The table names have now been amended, and there is no Table 3 in the revised manuscript

12. Please include captions for your Supporting Information files at the end of your manuscript, and update any in-text citations to match accordingly. 

Response: Please find all the captions for the supporting information at the end of the manuscript

We are grateful for your time and consideration of this manuscript. 

Please do let us know if you require any further clarification on any of the comments above.

Yours sincerely,

---

## [Decision Letter · Decision Letter 1]

14 Jul 2021

PONE-D-21-07362R1

High numbers of COVID-19 patients transit through non-COVID wards, and associated healthcare workers have high infection rates: An observational cross-sectional study

PLOS ONE

Dear Dr. Bowcock,

Thank you for submitting your manuscript to PLOS ONE. After careful consideration, we have decided that your manuscript does not meet our criteria for publication and must therefore be rejected.

Specifically:

ACADEMIC EDITOR: I also agree with the reviewer that Ethics committee approval will be needed for this study. 

I am sorry that we cannot be more positive on this occasion, but hope that you appreciate the reasons for this decision.

Yours sincerely,

Prasenjit Mitra, MD, MRSB, MIScT, FLS, FACSc, FAACC

Academic Editor

PLOS ONE

Reviewers' comments:

Reviewer's Responses to Questions

**Comments to the Author**

1. If the authors have adequately addressed your comments raised in a previous round of review and you feel that this manuscript is now acceptable for publication, you may indicate that here to bypass the “Comments to the Author” section, enter your conflict of interest statement in the “Confidential to Editor” section, and submit your "Accept" recommendation.

Reviewer #1: All comments have been addressed

2. Is the manuscript technically sound, and do the data support the conclusions?

Reviewer #1: Yes

3. Has the statistical analysis been performed appropriately and rigorously? 

Reviewer #1: Yes

4. Have the authors made all data underlying the findings in their manuscript fully available?

Reviewer #1: Yes

5. Is the manuscript presented in an intelligible fashion and written in standard English?

Reviewer #1: Yes

6. Review Comments to the Author

Reviewer #1: I feel that ethics committee approval is required for any study involving the collection of patient data and the analysis of patient samples, even if anonymised.

7. PLOS authors have the option to publish the peer review history of their article (what does this mean?). If published, this will include your full peer review and any attached files.

Reviewer #1: No

- - - - -

---

## [Author Response · Author response to Decision Letter 1]

1 Nov 2021

Dear Editors,

Thank you for reconsidering our manuscript which was provisionally accepted for publication in June 2021, but then rejected on a concern about ethics approval. This has been reviewed by a senior editor who is now satisfied that we followed correct procedures and can resubmit the manuscript.We now resubmit a clean version of the manuscript and one with tracked changes made since the original peer review. The changes are mainly to the methods section clarifying the ethics and consent processes.

We very much hope that you are now able to accept the manuscript as it has already passed peer review.

With many thanks,

---

## [Decision Letter · Decision Letter 2]

27 Jun 2022

PONE-D-21-07362R2High numbers of COVID-19 patients transit through non-COVID wards, and associated healthcare workers have high infection rates: An observational cross-sectional studyPLOS ONE

Dear Dr. Shee,

Thank you for submitting your manuscript to PLOS ONE. After careful consideration, we feel that it has merit but does not fully meet PLOS ONE’s publication criteria as it currently stands. Therefore, we invite you to submit a revised version of the manuscript that addresses the points raised during the review process.

Please see the reviewers' comments below, as well as the latest attachment, which contains some additional comments regarding presentation of information in tables. When revising your manuscript, please address in particular the comment regarding details of the multivariate analysis used in this study, and please note the request to provide the questionnaire used in this study as a Supplementary Information file.

We look forward to receiving your revised manuscript.

Kind regards,

Hugh Cowley

Staff Editor

PLOS ONE

Journal Requirements:

Additional Editor Comments (if provided):

Reviewers' comments:

Reviewer's Responses to Questions

**Comments to the Author**

1. If the authors have adequately addressed your comments raised in a previous round of review and you feel that this manuscript is now acceptable for publication, you may indicate that here to bypass the “Comments to the Author” section, enter your conflict of interest statement in the “Confidential to Editor” section, and submit your "Accept" recommendation.

Reviewer #1: All comments have been addressed

Reviewer #2: (No Response)

2. Is the manuscript technically sound, and do the data support the conclusions?

Reviewer #1: Yes

Reviewer #2: Yes

3. Has the statistical analysis been performed appropriately and rigorously? 

Reviewer #1: Yes

Reviewer #2: Yes

4. Have the authors made all data underlying the findings in their manuscript fully available?

Reviewer #1: Yes

Reviewer #2: Yes

5. Is the manuscript presented in an intelligible fashion and written in standard English?

Reviewer #1: Yes

Reviewer #2: Yes

6. Review Comments to the Author

Reviewer #1: This revised manuscript is well written and can be accepted for publication. The text in both Figures 1 and 2 are blurred.

Reviewer #2: The Methods section is presented in a single section. I recommend the authors to separate this section into specific sub-sections each one indicated by a subheading: for instance, subjects and place where the study was carried out, design of the study, survey, data analysis, ethics, etc.

The authors can add the questionnaire used as a supplementary material.

The authors mention that a multivariate analysis was used, though they did not include any detail regarding the statistical tool used for this purpose. Please elaborate more on this and provide the approach used for such multivariate analysis.

7. PLOS authors have the option to publish the peer review history of their article (what does this mean?). If published, this will include your full peer review and any attached files.

Reviewer #1: No

Reviewer #2: **Yes:**

---

## [Author Response · Author response to Decision Letter 2]

7 Aug 2022

Thank you for giving us the opportunity to revise this manuscript. We have addressed all the points made by the reviewers.

1. Figures 1 and 2 which were blurred have now been uploaded in a different format and are now clearer.

2. The methods section has now been broken up into sub-sections as suggested by the reviewer. The text has not changed substantially but has been rearranged. The questionnaire has been uploaded in the supplementary material. Detail about the statistical tool used and the approach to the multivariate analysis has been added in the ‘Analysis’ section.

3. All changes to the tables as highlighted in the attached document, have been completed. 

We now hope that our manuscript fully meets your requirements.

---

## [Decision Letter · Decision Letter 3]

13 Sep 2022

High numbers of COVID-19 patients transit through non-COVID wards, and associated healthcare workers have high infection rates: An observational cross-sectional study

PONE-D-21-07362R3

Dear Dr. Bowcock,

We’re pleased to inform you that your manuscript has been judged scientifically suitable for publication and will be formally accepted for publication once it meets all outstanding technical requirements.

Kind regards,

Academic Editor

PLOS ONE

Additional Editor Comments (optional):

Reviewers' comments:

Reviewer's Responses to Questions

**Comments to the Author**

1. If the authors have adequately addressed your comments raised in a previous round of review and you feel that this manuscript is now acceptable for publication, you may indicate that here to bypass the “Comments to the Author” section, enter your conflict of interest statement in the “Confidential to Editor” section, and submit your "Accept" recommendation.

Reviewer #1: All comments have been addressed

Reviewer #3: (No Response)

2. Is the manuscript technically sound, and do the data support the conclusions?

Reviewer #1: Yes

Reviewer #3: Yes

3. Has the statistical analysis been performed appropriately and rigorously? 

Reviewer #1: Yes

Reviewer #3: I Don't Know

4. Have the authors made all data underlying the findings in their manuscript fully available?

Reviewer #1: Yes

Reviewer #3: Yes

5. Is the manuscript presented in an intelligible fashion and written in standard English?

Reviewer #1: Yes

Reviewer #3: Yes

6. Review Comments to the Author

Reviewer #1: All the reviewers' comments have been addressed satisfactorily. The figures, although still a little blurred, are acceptable. The manuscript may be accepted for publication.

Reviewer #3: The article confirms a known phenomenon, healthcare workers who come into contact with COVID-19 positive patients are at greater risk of infection than others.

7. PLOS authors have the option to publish the peer review history of their article (what does this mean?). If published, this will include your full peer review and any attached files.

Reviewer #1: **Yes: **

Reviewer #3: **Yes: **

---

## [Editor Report · Acceptance letter]

23 Sep 2022

PONE-D-21-07362R3 

High numbers of COVID-19 patients transit through non-COVID wards, and associated healthcare workers have high infection rates: An observational cross-sectional study 

Dear Dr. Bowcock:

I'm pleased to inform you that your manuscript has been deemed suitable for publication in PLOS ONE. Congratulations! Your manuscript is now with our production department. 

Kind regards, 

on behalf of

Dr. Robert Jeenchen Chen 

Academic Editor

PLOS ONE